# A randomized trial: The safety, pharmacokinetics and preliminary pharmacodynamics of ropivacaine oil delivery depot in healthy subjects

**Wu-dang Lu[1,2], Dan-ling Zhao[3], Mei-xia Wang[4], Ya-qi Jiao[2], Ping Chi[4], Min Zhang[3], Bo Ma[3], Jian-ping Dong[1], Hai-bo Zhang[3], Yi Yang[1], Ye Tian[3], Min-quan Hui[1], Bo Yang[3], Yong-xiao Cao🄳[1] ***

**1** School of Basic Medical Sciences, Xi'an Jiaotong University Health Science Center, Xi'an, Shaanxi, China, **2** Xi'an Libang Pharmaceutical Co., Ltd, Xi'an, Shaanxi, China, **3** Beijing Aicomer Pharmaceutical Technology Co., Ltd., Beijing, China, **4** Beijing You'an Hospital, Capital Medical University, Beijing, China

* yxy@xjtu.edu.cn

## Abstract

### Introduction

Ropivacaine oil delivery depot (RODD) can slowly release ropivacaine and block nerves for a long timejavascript:;. The aim of the present work was to investigate the safety, pharmacokinetics, and preliminary pharmacodynamics of RODD in subcutaneous injection among healthy subjects.

### Methods

The abdomens of 3 subjects were subcutaneously administered with a single-needle RODD containing 12~30 mg of ropivacaine. The irritation, nerve blocking range and optimum dose were investigated. Forty-one subjects were divided into RODD groups containing 150, 230, 300, 350 and 400 mg of ropivacaine and a ropivacaine hydrochloride injection (RHI) 150 mg group. Multineedle subcutaneous injection of RODD or RHI was performed in the abdomens of the subjects. The primary endpoint was a safe dose or a maximum dose of ropivacaine (400 mg). Subjects' vital signs were observed; their blood was analyzed; their cardiovascular system and nervous systems were monitored, and their dermatological reactions were observed and scored. Second, the ropivacaine concentrations in plasma were determined, pharmacokinetic parameters were calculated, and the anesthetic effects of RODD were studied, including RODD onset time, duration and intensity of nerve block.

### Results

Single-needle injection of RODD 24 mg was optimal for 3 subjects, and the range of nerve block was 42.5±20.8 mm. Multineedle subcutaneous injection of RODD in the abdomens of subjects was safe, and all adverse events were no more severe than grade II. The incidence rate of grade II adverse events, such as pain, and abnormal ST and ST-T segment changes

**Data Availability Statement:** All relevant data are within the paper and its Supporting Information files.

**Funding:** The author(s) received no specific funding for this work.

**Competing interests:** The authors have declared that no competing interests exist.

on electrocardiography, was approximately 1%. The incidence rate of grade I adverse events, including erythema, papules, hypertriglyceridemia, and hypotension was greater than 10%. Erythema and papules were relieved after 24 h and disappeared after 72 h. Other adverse reactions disappeared after 7 days. The curve of ropivacaine concentration-time in plasma presented a bimodal profile. The results showed that ropivacaine was slowly released from the RODD. Compared with the 150 mg RHI group, $T_{max}$ was longer in the RODD groups. In particular, $T_{max}$ in the 400 mg RODD group was longer than that in the RHI group (11.8±4.6 h vs. 0.77±0.06 h). The $C_{max}$ in the 150 mg RODD group was lower than that in the 150 mg RHI group (0.35±0.09 vs. 0.58±0.13 µg·mL$^{-1}$). In particular, the $C_{max}$ increased by 48% when the dose was increased by 2.6 times in the 400 mg group. $C_{max}$, the AUC value and the intensity of the nerve block increased with increasing doses of RODD. Among them, the 400 mg RODD group presented the strongest nerve block (the percentage of level 2 and 3, 42.9%). The corresponding median onset time was 0.42 h, and the duration median was 35.7∼47.7 h.

## Conclusions

RODD has a sustained release effect. Compared with the RHI group, $T_{max}$ was delayed in the RODD groups, and the duration of nerve block was long. No abnormal reaction was found in the RODD group containing 400 mg of ropivacaine after subcutaneous injection among healthy subjects, suggesting that RODD was adequately safe.

## Trial registration

Chictr.org: CTR2200058122; Chinadrugtrials.org: CTR20192280.

## Introduction

Ropivacaine is an S-enantiomer amide local anesthetic with high pKa and low lipid solubility. Its nerve block effects are similar to those of bupivacaine, but it has lower neurotoxicity, cardiovascular toxicity, and local muscular toxicity [1–4].

At present, ropivacaine hydrochloride injection is widely used in the clinic. Commonly, the injection is administered subcutaneously at the surgical incision or injected directly at the ganglion for preemptive analgesia before the operation [5, 6]. During operation of childbirth, ropivacaine is combined with a low dose of opioid for anesthesia maintenance [7, 8]. Moreover, ropivacaine is used in a variety of methods for postoperative analgesia. 1) A mixture of ropivacaine and opioids is administered for persistent nerve block analgesia by an epidural analgesia pump [9]. 2) Ropivacaine solution is administered subcutaneously at the incision, which is suitable for large-incision operations with dense cortical nerve distribution, such as in the chest, abdomen and limbs [10–13]. 3) Ropivacaine is sprayed intraperitoneally to reduce pulling pain, and is often used in digestive system surgery, such as stomach, liver, gallbladder, urinary system and gynecological uterus- and ovary-related laparoscopic surgery [14–18]. This postoperative analgesic management can effectively relieve postoperative pain in patients, reduce the amounts of prescribed opioids, and promote postoperative recovery.

After single administering ropivacaine hydrochloride solution to the surgical incision, a nerve block is produced; the analgesic duration is less than 8 h, which likely will not meet the needs of postoperative analgesia (i.e., more than 24 h) [19, 20]. Although the increased dose of

ropivacaine can prolong the duration of the nerve block, the increase in plasma drug concentration will significantly increase the incidence of related neurotoxicity and cardiovascular toxicity [21]. Therefore, there is obvious clinical significance in prolonging the postoperative nerve block time of ropivacaine and reducing adverse reactions.

To date, more than three slow-release formulations of ropivacaine have been studied. Ropivacaine liposome gels containing infrared photosensitizers and phospholipids can be retained *in vivo* for 10 days [22]. Poloxamer 407 has been used as an excipient to prepare a ropivacaine-loaded gel to treat pain after thoracoscopic pneumonectomy. The analgesic effect was comparable to that of continuous infusion of 0.2% ropivacaine for 48 h [23]. Poloxamer 407 and polycaprolactone electrospun fibers have been used as excipients to prepare ropivacaine hydrogels that blocked rat nerves for more than 30 h [24]. However, the safety of new excipients has yet to be fully elucidated. Our preliminary formulation and technology study showed that liposomes are at risk of sudden drug release due to incomplete encapsulation of the drug, and the drugs in the gel tend to be released in large quantities because of their water solubility. In contrast, drugs and major excipients of oil-soluble preparations are hydrophobic, thus avoiding the above problems.

Ropivacaine is insoluble in water and is slowly released from the oil solution. Castor oil and phospholipids were used as carriers to design a ropivacaine suspension formula that could sustainably block nerves for 36 h [25]. We disclosed the dogs' pharmacokinetic data and preliminary toxicity of ropivacaine in soybean oil solution [26]. Compared with ropivacaine hydrochloride injection, the $C_{max}$ value of the ropivacaine oil delivery system was decreased by 80%, and $T_{max}$ was significantly delayed. The dog was given a subcutaneous preparation of 60 mg·kg$^{-1}$. There was no significant change in the dog's behaviors, blood count, coagulation, blood and urine biochemistry. The ropivacaine oil delivery depot has not been found to confer significant toxicity [26].

Currently, our ropivacaine oily delivery depot (RODD) has been approved for phase I clinical research by the Center for Durg Evaluation of National Medical Products Administration of China. This study evaluated the safety, pharmacokinetics and preliminary efficacy of RODD in healthy subjects.

## Materials and methods

### Materials

Ropivacaine was obtained from Libang Pharmaceutical Co., Ltd., China. The purity was 99%. Benzyl alcohol and benzyl benzoate were purchased from Hunan Er-kang Pharmaceutical Co., Ltd., China. Soybean oil was produced by Tieling Pharmaceutical Co., Ltd., China. Ropivacaine hydrochloride injection (RHI) was manufactured by AstraZeneca AB, Sweden. The specifications are 100 mg:10 mL. RODD was composed of 2% ropivacaine, 4% benzyl alcohol, 30% benzyl benzoate and approximately 64% soybean oil and was commissioned to be produced at the GMP (Good Manufacturing Practice)-certified plant of Jiangsu Taikang Biomedical Co., Ltd, batch number CR20191001. Ropivacaine-D7, a reference substance, was obtained from the United States Pharmacopoeia.

### RODD preparation

Ropivacaine (200 mg) was added to benzyl alcohol. After the ropivacaine was dissolved, benzyl benzoate and soybean oil were added to the ropivacaine solution up to 10 mL. Then, the mixture was heated and stirred until the solution was clear. After being filtered and sterilized with a filter, the bulk solution, i.e., RODD, was obtained, divided into vials, filled with nitrogen and

sealed. The concentration of RODD was 20 mg·mL$^{-1}$, and the stability of RODD was more than 2 years at 2~8˚C.

## Selection of subjects

The inclusion criteria included an age range of healthy subjects of 18~50 years for men or women and a body mass index ranging from 19.0 to 26.0 kg·m$^{-2}$.

The exclusion criteria included subjects with abnormal vital signs; clinical/laboratory tests (blood routine, urine routine and blood biochemical, serology, electrocardiogram (ECG)), cerebrovascular, liver, kidney, respiratory, blood, endocrine, immune, mental, or gastrointestinal diseases; porphyria, malnutrition or blood diseases; scar constitution; analgesic use in the past 2 weeks; long-term drinking of tea and coffee, blood donation within the past three months; and participation in other drug clinical trials within the past three months.

A total of 50 healthy subjects were selected, with no restriction regarding gender. They were informed about the purpose, procedure, possible benefits and risks of the study and signed the informed consent form.

## Ethics statement

The research was designed and conducted in accordance with the current "Good Clinical Practice (GCP) of China", "The International Council for Harmonization of Technical Requirements for Pharmaceuticals for Human Use (ICH)-GCP", "Guidelines for Ethical Review of Drug Clinical Trials" and "Declaration of Helsinki". Clinical trial protocol revision and implementation were approved and supervised by the Hospital Ethics Committee. The protocol of the clinical study was approved by the Ethics Committee of Beijing You'an Hospital affiliated with Capital Medical University on September 5, 2019. The study recruited the first subject on December 2, 2019, and enrolled the last participant on June 25, 2021.

The study has completed registration in the drug clinical trial registration and information disclosure platform, which is the legal registration platform for clinical information used by the drug evaluation center of the state medical products administration of China. The clinical trial record number is CTR20192280 in China. At present, the study has completed registration in the Chinese Clinical Trial Registry (ChiCTR), and the clinical trial record number is ChiCTR2200058122.

## Trial design

Three subjects received single-needle subcutaneous injection of different doses of RODD in the abdomen to determine the optimal dose, anesthesia range, and irritation for the subsequent multiple-needle subcutaneous injection (**study 1**). Forty-one subjects underwent multineedle subcutaneous injection into abdomens with an escalating dose of RODD. The primary endpoint was safety, defined as the occurrence of grade III adverse events, or the attainment of the maximum dose of RODD (400 mg of ropivacaine). Additionally, the pharmacokinetics and preliminary effects were studied (**study 2**). The grouping and test process is shown in Fig 1.

## Study 1: Single-needle subcutaneous injection exploration

Three subjects were selected for a single-needle subcutaneous injection to determine the optimal dose. Fig 2A shows that the abdomen of each subject was divided into six regions from the ensisternum and marked I to VI clockwise. One of these areas was randomly selected (using a random number table on the computer) as the starting point, and the areas were marked 1~5

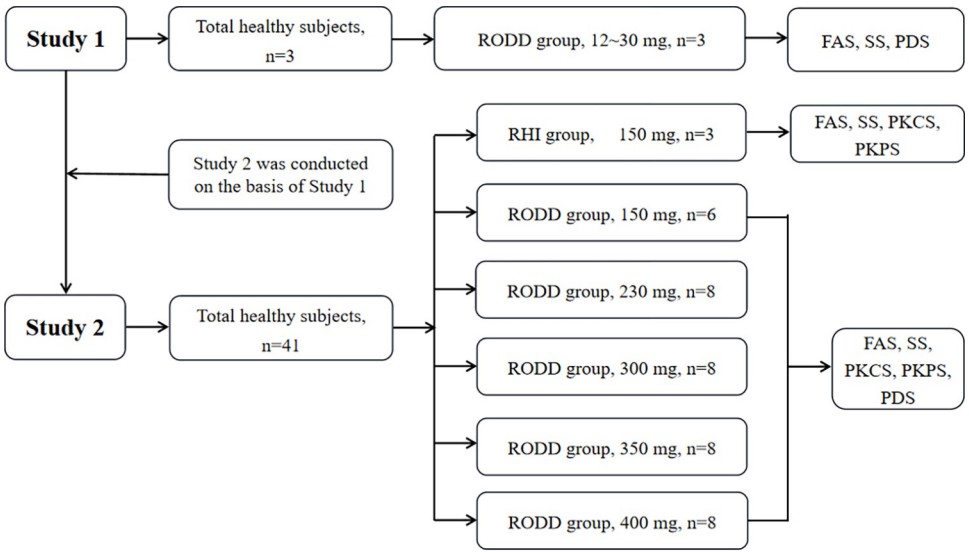

FAS: Full analysis set; SS: Safety set; PKCS: Pharmacokinetic concentrations set; PKPS: Pharmacokinetic parameters set; PDS: Pharmacodynamics set; RODD: Ropivacaine oil delivery depot; RHI: Ropivacaine hydrochloride injection.

**Fig 1. Grouping and flow chart of healthy subjects' abdominal subcutaneous injection of RODD.**

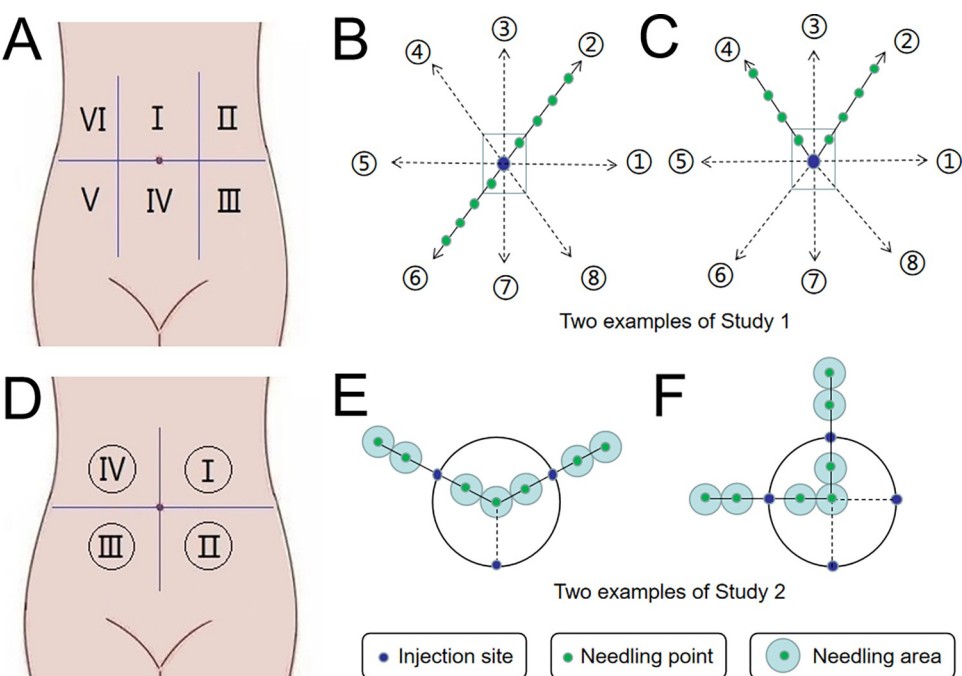

**Fig 2. Schematic diagram of healthy subjects' abdominal subcutaneous injection of RODD.** A and D are the distribution designs of single-needle and multineedle injection areas under the abdomen skin, which were areas I~ and I~IV, respectively. B and C are examples of single-needle injection points and needle test points; E and F are examples of multineedle injection points and acupuncture test points on a circle, respectively.

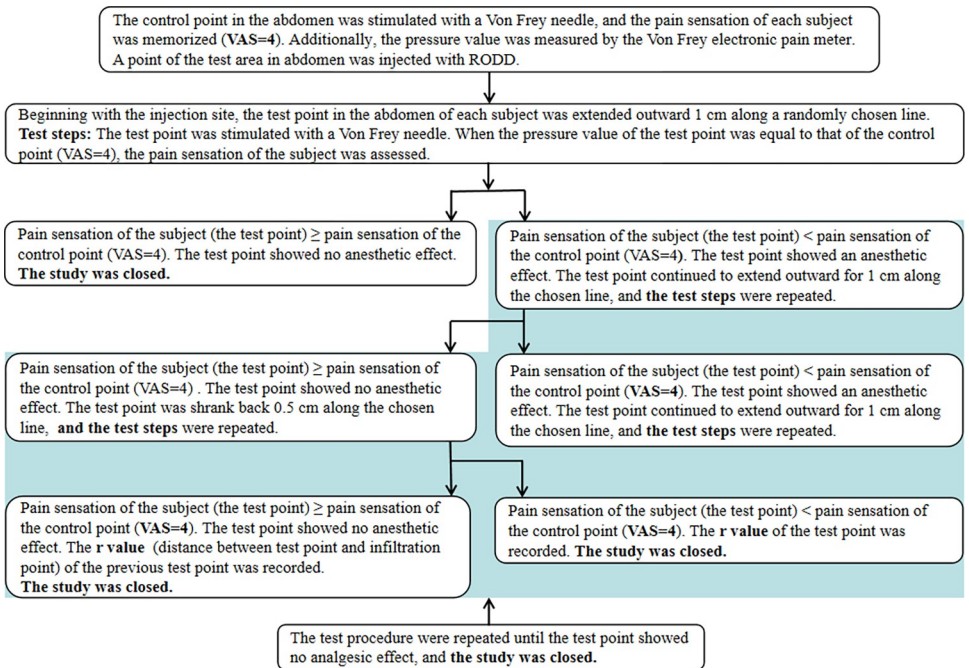

**Fig 3. The evaluative process of analgesia range in single-needle abdominal subcutaneous injection with RODD in healthy subjects.**

clockwise. The injection doses of RODD (containing ropivacaine) were 12 mg (0.6 mL), 16 mg (0.8 mL), 20 mg (1.0 mL), 24 mg (1.2 mL) and 30 mg (1.5 mL). Area 6 was used as the blank control. An interval of 1 h between two adjacent doses was used to observe any adverse reactions. This experiment was based on the VAS score [27]. The range of the nerve block was measured through acupuncture by a Von Frey electronic pain meter (IITC Life Science) before and 0.25, 0.5, 0.75, 1, 1.5, 2, 3, 4, 5, 6, 7, 8, 9, 10, 11, and 12 h after subcutaneous administration.

The test methods are shown in Fig 2B and 2C. The injection sites in the abdominal region were taken as the starting point, eight test directions were designed, and two lines (r1 and r2) were randomly selected to test the nerve block effect in 8 directions around the injection site that started from the injection site. Test points were set along the line from the injection site, and the distance between two points was 1 cm. The test steps are shown in Fig 3. The nerve block range (r1+r2) at each injection site was recorded, and the area under the block range-time curve was calculated to determine the optimal infiltration dose.

At 0.5 h before administration, and at 0.3, 0.6, 1, 2, 6, 24, and 72 h after administration, the symptoms of injection sites were recorded, and irritation was scored.

## Study 2: Multiple-needle subcutaneous injection

Forty-one subjects were selected to receive the multineedle subcutaneous injection and were randomly divided into the 150, 230, 300, 350 and 400 mg RODD groups (n = 6~8) and the ropivacaine hydrochloride injection (RHI) 150 mg group (n = 3). Each subject received a subcutaneous injection of RODD or RHI. The abdomen was divided into four areas by drawing horizontal and vertical lines through the navel, as shown in Fig 2D, marked as I, II, III and IV in a clockwise direction.

One area was randomly selected and marked as 1, while the other areas were marked clockwise as 2~4. Areas 1~3 were administered with RODD, and area 4 was the blank control for the nerve block test. A point in each region was selected as the center of the circle (the distance between the two centers was more than 8 cm), and the optimal block range and dose were obtained via the single-needle subcutaneous injection in **study 1**. The number of needles in each dose group was calculated as follows:

Number of needles in the group = total dose of the group/optimal dose.

Number of needles on circle 1 or 2 = the number of needles on each group/3.

The remaining needles were allocated to the third circle. The needles on the circumference were evenly distributed. The intensity of the nerve block in the abdominal injecting area was measured in all subjects at different times after administration. The subjects were hospitalized for 72 h and followed up for 7 days after discharge.

As shown in Fig 2E and 2F, two injection sites on a circle were randomly selected, which were connected to the center of the circle and extended outward 2 cm. The four test points on the straight line are the center of the circle, 1 cm away from the center of the circle (inside the circle), and 1 and 2 cm away from the injection site (outside the circle). The test area was within 0.5 cm around the test point. The VAS was used to assess pain intensity via acupuncture in any two directions of circles 1 and 2 [27], and each point or area was tested once. The evaluation process of nerve block intensity is shown in Fig 4.

## Safety assessment

Vital signs such as blood pressure, pulse, respiration and oxygen saturation were measured in the supine position for the subjects 1 h before and 1, 2, 4, 8, 12, 24, 48 and 72 h after administration. Routine blood tests, electrocardiograms, blood biochemistry and urine tests were performed 24 h before and 72 h after administration. The symptoms of injection sites were

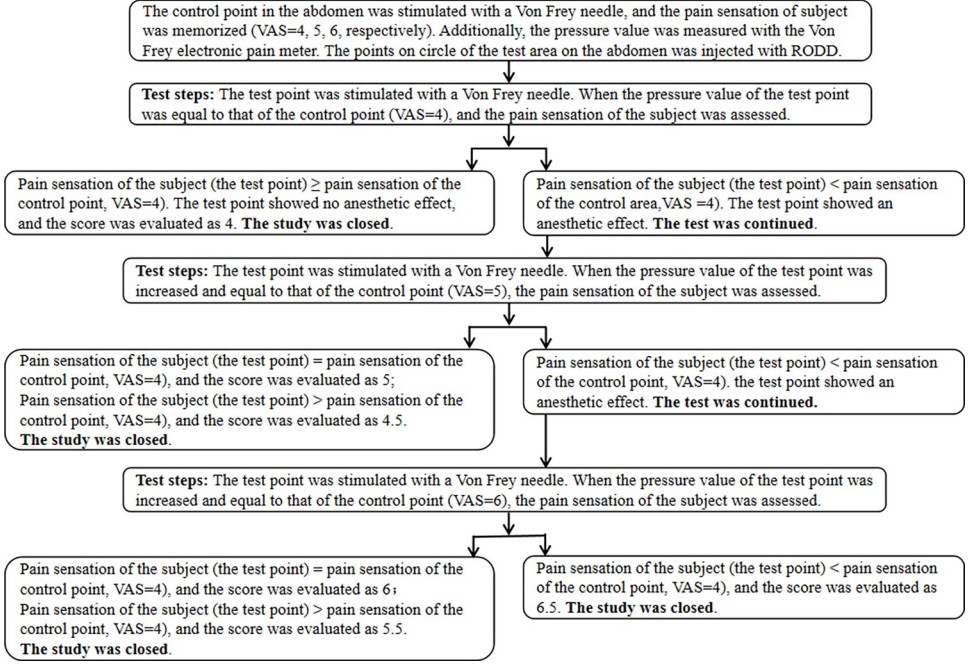

**Fig 4. Scoring process of analgesia intensity for subcutaneous multineedle injection of RODD in the abdomens of healthy subjects in Study 2.**

**Table 1. The levels of local skin reaction in adverse events.**

| Level | Dermatological reaction |
|---|---|
| 0 | No irritation was observed |
| 1 | A slight erythema can just be observed |
| 2 | Obvious erythema, mild edema or papules were observed |
| 3 | Erythema and papules are prominent |
| 4 | Edema was significant |
| 5 | Erythema, edema and papules |

recorded, and irritation was assessed at 0.5 h before and 0.3, 0.6, 1, 2, 6, 24 and 72 h after administration. The criteria for local adverse events are shown in Table 1. The safety evaluation was based on the Evaluation Criteria for "Common Adverse Events" published by Chinese Clinical regulatory authorities (V5.0, 2017.11). Routine blood tests included red blood cell count, white blood cell count, hemoglobin, hematocrit, platelet count, eosinophil count, basophil count, neutrophil count, lymph cell count and monocyte count. Routine urine examination items included glucose, ketone bodies, protein, urinary occult blood, white blood cells, bilirubin, nitrite, pH value, urochologen and specific gravity. Blood biochemical indices included potassium, sodium, chlorine, calcium, magnesium, phosphorus, blood glucose, creatinine, urea, glutamine transferase, uric acid, lactate dehydrogenase, total bilirubin, aspartic ammonia acid aminotransferase, alanine aminotransferase, alkaline phosphatase, triglyceride, total cholesterol, total protein and albumin.

## Pharmacokinetics

The liquid chromatography coupled with tandem mass spectrometry (LC–MS/MS) analysis method of ropivacaine was optimized on the basis of previous reports [28–30]. The liquid chromatography analysis conditions are shown in Table 2, including a column temperature of 20˚C, injection volume of 10 μL, flow rate of 0.40 mL·min$^{-1}$, and sample chamber temperature of 2~8˚C. Mass spectrometry conditions included an API 4000 mass spectrometer, Turbo Ionspray (APCI$^+$) ion source, MRM detection mode, GS1 and 2 of 60 psi, TEM: 500˚C, CUR: 30 psi, CAD: 5 psi, ropivacaine ion pair: 275.1/126.1, scanning time 200 ms, DP: 60 V, EP: 10 V,

**Table 2. Liquid chromatography tandem-mass spectrometry quantitative liquid phase analysis conditions.**

| / | Conditions | | |
|---|---|---|---|
| Instrument | Agilent 1100 High phase liquid chromatograph (degassing machine G1322A, binary pump G1312A, column temperature chamber G1316A) Automatic sampler (CTC Analytics HTC PAL) Chromatographic column (Hypersil GOLD, 2.1·50 mm, 5 μm) | | |
| Mobile phase | A: Acetonitrile-water-formic acid (10:90:0.2, v/v/v) | | |
| | B: Acetonitrile-water-formic acid (90:10:0.2, v/v/v) | | |
| Gradient elution condition | Time (min) | A% | B% |
| | 0.00 | 90.0 | 10.0 |
| | 0.50 | 90.0 | 10.0 |
| | 0.70 | 5.0 | 95.0 |
| | 2.20 | 5.0 | 95.0 |
| | 2.21 | 0.0 | 100.0 |
| | 2.60 | 0.0 | 100.0 |
| | 2.61 | 90.0 | 10.0 |
| | 4.00 | 90.0 | 10.0 |

CE: 24 V; CXP: 13 V. The extraction and detection method were verified, and the content of ropivacaine in plasma samples was determined.

At 0.25 h before and 0.5, 1, 1.5, 2, 4, 6, 8, 10, 12, 14, 21, 24, 27, 30, 33, 36, 48 and 72 h after administration, 4 mL of venous blood was drawn into a blood collection tube containing EDTA-$K_2$ anticoagulant and centrifuged at $2000 \times g$ for 10 min. The upper layer of plasma was transferred into 2-mL cryopreserved tubes (Axygen, made in USA) and then stored in a refrigerator (-60~-90˚C). The plasma was treated with the protein precipitation method before analysis. Specifically, the plasma was thawed at room temperature and blended by vortexing, 50 μL of plasma was transferred into a centrifuge tube, and 160 μL of ropivacaine-D7 acetonitrile solution (internal standard ropivacaine-D7 15 ng·$mL^{-1}$) was added. After 1 min of vortexing, the sample was centrifuged at 12 000 rpm for 10 min, and 40 μL supernatant was transferred into the sample tube. Then, 160 μL ultrapure water was added to the same tube and vortexed for testing. The injection volume was 10 μL for quantitative analysis by LC–MS/MS.

### Preliminary pharmacodynamics

At 0.25 h before and 0.25, 1, 4, 8, 10, 12, 14, 18, 21, 24, 27, 30, 33, 36, 48 and 72 h after administration, the analgesia intensity was measured by acupuncture on circles 1 and 2 of the abdomen, and the onset time and effective duration were evaluated. The distance between two adjacent injection sites was recorded.

### Statistical analysis

All data were analyzed by SAS software (version 9.4), and the indicators of each group were compared with baseline. The ropivacaine concentration-time curves were drawn, and pharmacokinetic parameters were calculated (Phoenix Win Nonlin 8.2). A fitting power model was used to evaluate the dose-linear relationship of pharmacokinetic parameters $C_{max}$, $AUC_{0-t}$ and AUC0-∞. Pharmacokinetic parameters transformed by natural logarithm were used as dependent variables, and the natural logarithm transformed value ln(dose) of the dose group was used as an independent variable. Then, a linear model was fitted. Estimates of ln(dose) regression coefficients and their 90% confidence intervals were calculated.

Local irritation scores of RODD were evaluated by descriptive statistics, and the frequency and percentage of categorical or ordinal variables were summarized. The mixed effect model was used to analyze the changes in acupuncture pain intensity at different time points before and after administration. In the model, changes in pain intensity at different time points in different administration areas compared with baseline were dependent variables. Pain intensity before administration was a covariate. Administrative area/blank area, grouping and visit point were fixed effects. The onset time and duration of RODD nerve block were compared by median (95%, CI). The analgesic intensity was calculated as a percentage, and the range of nerve block and distance adjacent to injection sites were expressed as the mean±SD.

## Results

### Subject disposition and demographics

A total of 44 healthy subjects (75% men and 25% women), with an average age of 29.3±7.1 years were enrolled. Han nationality accounted for 93% and other ethnic groups accounted for 7%. The average height was 168.6±8.4 cm, the average weight was 63.9±9.7 kg, and the average body mass index was 22.3±1.9 kg·$m^{-2}$. The specific demographic data of healthy subjects are shown in Table 3. In addition, whole blood and blank plasma samples from 6 healthy subjects were collected for the methodology and stability study of the required biological samples.

**Table 3. Demographics and baseline characteristics (mean±SD).**

| Group | | Study 1 | | Study 2 | | | | | |
|---|---|---|---|---|---|---|---|---|---|
| | | Ropivacaine oil delivery depot | Ropivacaine hydrochloride injection | Ropivacaine oil delivery depot | | | | | |
| Dose (mg) | | 12∼30 | 150 | 150 | 230 | 300 | 350 | 400 | |
| | n | 3 | 3 | 6 | 8 | 8 | 8 | 8 | |
| Age | Average | 28.7±4.1 | 28.3±2.9 | 26.7±6.3 | 28.1±8.1 | 31.3±10.0 | 26.3±4.5 | 33.9±5.1 | |
| | Min, Max | 25, 33 | 25, 30 | 20, 38 | 19, 43 | 21, 45 | 20, 32 | 28, 43 | |
| Sex n (%) | Male | 2 (66.7) | 3 (100.0) | 5 (83.3) | 5 (62.5) | 6 (75.0) | 6 (75.0) | 6 (75.0) | |
| | Female | 1 (33.3) | 0 (0.0) | 1 (16.7) | 3 (37.5) | 2 (25.0) | 2 (25.0) | 2 (25.0) | |
| Nation n (%) | Ethnic han | 2 (66.7) | 3 (100.0) | 6 (100.0) | 8 (100.0) | 6 (75.0) | 8 (100.0) | 7 (87.5) | |
| | other | 1 (33.3) | 0 (0.0) | 0 (0.0) | 0 (0.0) | 2 (25.0) | 0 (0.0) | 1 (12.5) | |
| High (cm) | Average | 164.3±8.1 | 171.3±12.2 | 171.5±6.0 | 169.9±8.1 | 166.9±10.1 | 166.3±7.2 | 168.2±9.6 | |
| | Min, Max | 157.0, 173.0 | 158.0, 182.0 | 164.0, 179.0 | 157.5, 183.0 | 150.0, 183.0 | 157.5, 179.0 | 148.0, 179.0 | |
| Weight (kg) | Average | 61.1±4.4 | 64.8±9.8 | 67.4±10.6 | 65.4±11.1 | 62.5±10.1 | 61.2±10.2 | 63.3±8.8 | |
| | Min, Max | 57.7, 66.1 | 54.9, 74.5 | 55.1, 80.9 | 50.0, 80.5 | 47.8, 79.4 | 47.9, 77.4 | 50.9, 74.4 | |
| BMI (kg/m$^2$) | Average | 22.6±1.3 | 22.0±0.5 | 22.8±2.5 | 22.5±2.1 | 22.3±2.0 | 22.0±1.9 | 22.3±2.1 | |
| | Min, Max | 21.7, 24.1 | 21.5, 22.5 | 19.5, 25.6 | 20.2, 25.3 | 20.0, 25.3 | 19.3, 24.9 | 19.4, 24.7 | |

BMI, body mass index; SD, standard deviation; Min, minute; Max, maximum.

## Optimal dose of single-needle injection of RODD

The block range-time curve after single-needle subcutaneous injection of RODD is shown in Fig 5A. In the 24 mg and 30 mg RODD groups, the nerve block range (i.e., average diameter, 42.5±20.8 and 46.2±24.3 mm) and the areas under the curve (574±228 and 573±56 h·mm) were higher than those in the 12, 16 and 20 mg groups (Table 4). The anesthesia duration was maintained for more than 12 h. Slight erythema around the injection site was alleviated after 2 h, with an incidence of 28%, and disappeared after approximately 24 h (Fig 5B). The local irritation of RODD at the 24 mg injection site was slightly lower than that at the 30 mg injection site. It was suggested that the optimal dose was 24 mg with a block range of 40 mm.

## Safety of multiple-needle injection of RODD

The abdomen of the subjects was divided into 4 regions, and a circle with a diameter of 40 mm was marked in every region. The number of injection sites and drug quantity distribution on the circle are shown in Table 5. The incidences of adverse reactions of subjects in the 150, 230, 300, 350, 400 mg RODD, and 150 mg RHI groups were 83.3% (5/6), 75.0% (6/8), 100.0% (8/8), 62.5% (5/8), 87.5% (7/8) and 100.0% (3/3), respectively. Common grade I adverse events included erythema at the infiltrating site (40.9%, 18/41), papules (15.9%, 7/41), hypertriglyceridemia (25.0%, 11/41), and hypotension (20.4%, 9/41). There was no correlation between the incidence of adverse events and doses. Erythema and papules are shown in Fig 6. Grade II adverse events mainly included ST segment depression and ST segment abnormality (2.3%, 1/41). There were no grade III or above adverse events (Table 6). Most of the erythema and papules in the injecting site were relieved 24 h after administration and almost disappeared within 72 h. Blood pressure and ECG returned to normal 24 h after administration. The data suggested that RODD has adquate safety.

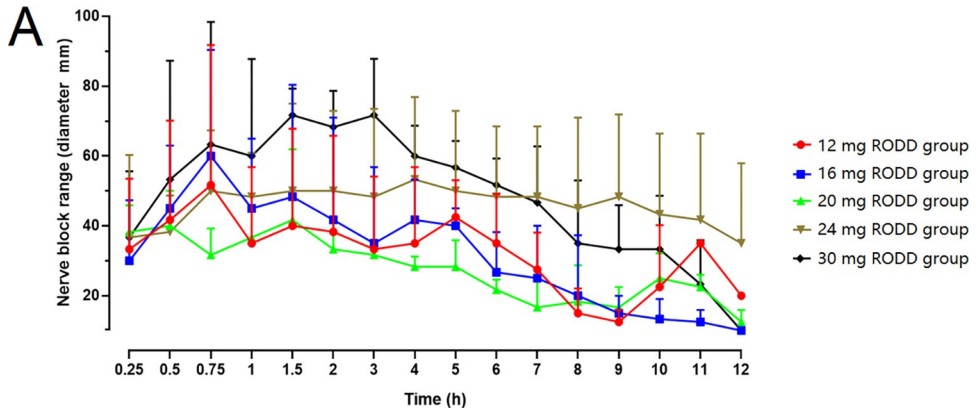

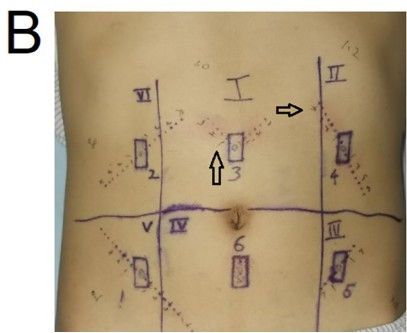

**Fig 5. Curves of nerve block range-time after subcutaneous injection of 12~30 mg RODD (mean±SD, A).**
Irritation after subcutaneous injection of RODD for 2 h on the abdomen of the subject (B), and slight erythema appeared on the skin around some injection sites (arrows).

**Table 4. Analgesic range diameter and area under the time curve of single-needle abdominal subcutaneous injection of ropivacaine oil delivery depot (mean±SD, n = 3).**

| Group | Ropivacaine oily delivery depot | | | | |
|---|---|---|---|---|---|
| Dose (mg) | 12 | 16 | 20 | 24 | 30 |
| Analgesic range diameter (mm) | 31.8±20.8 | 31.8±20.9 | 27.4 ±11.6 | 42.5 ±20.8 | 46.2±24.3 |
| AUC (h·mm) | 304.2 ±229.9 | 338.4±52.1 | 286.7 ±29.0 | 574.2 ±227.6 | 572.8 ±55.7 |

AUC, area under the curve; SD, standard deviation.

**Table 5. Circumferential distribution of multineedle injection of ropivacaine oil delivery depot in abdomens of the subjects.**

| Group | Ropivacaine hydrochloride injection | Ropivacaine oil delivery depot | | | | |
|---|---|---|---|---|---|---|
| Dose (mg) | 150 | 150 | 230 | 300 | 350 | 400 |
| Volum (mL) | 17 | 7.5 | 11.5 | 15 | 17.5 | 20 |
| n | 3 | 6 | 8 | 8 | 8 | 8 |
| Circle 1[1], Circle 2[1] | 1.2 mL×5 | 1.2 mL×2 | 1.2 mL×3 | 1.2 mL×4 | 1.2 mL×5 | 1.2 mL×6 |
| Circle 3 | 1.2 mL×4, 0.2 mL×1 | 1.2 mL×2, 0.3 mL×1 | 1.2 mL×3, 0.7 mL×1 | 1.2 mL×4, 0.6 mL×1 | 1.2 mL×4, 0.7 mL×1 | 1.2 mL×4, 0.8 mL×1 |

RODD, ropivacaine oil delivery depot; RHI, ropivacaine hydrochloride injection. 150 mg RHI, n = 3; 150 mg RODD, n = 6; 230～400 mg RODD, n = 8, respectively.
Note
[1], RODD or RHI given on circumference 1 and 2, respectively.

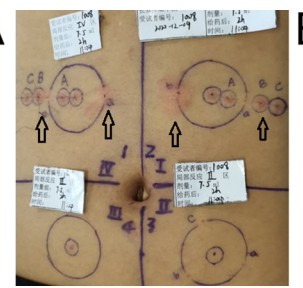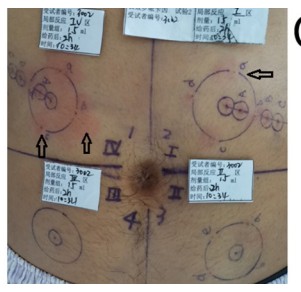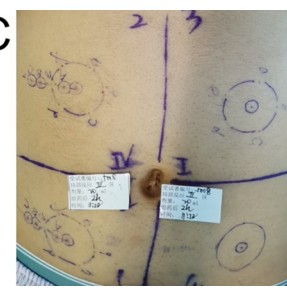

**Fig 6. Local dermatological reactions (arrows) of healthy subjects in Study 2 at 2 h after subcutaneous injection of RODD.** Subjects A, B, and C were healthy subjects. Subject A showed mild erythema around the injection site in the 150 mg RODD group. Subject B showed erythema and mild papules around the injection site in the 300 mg RODD group, and Subject C did not show any abnormalities around the injection site in 400 mg RODD group.

## Methodological verification

The analysis method of ropivacaine had superior selectivity. The standard curve range was 2.5~2000 ng·mL$^{-1}$, the correlation coefficient was greater than 0.998, the inter batch precision (CV%) was 0.81%~2.05%, the inter batch accuracy (RE%) was -2.45%~2.06%, and the lowest limit of quantification was 2.5 ng·mL$^{-1}$. Quality control samples were prepared by adding different doses of ropivacaine to blank plasma, and then the samples were processed according to the sample handling method. The recovery rates of the quality control samples (the quality control of low (7.5 ng·mL$^{-1}$), medium (150 ng·mL$^{-1}$) and high (1.5 mg·mL$^{-1}$) concentrations) were 102.1%±4.5%, 102.8%±1.5% and 103.1%±1.9%, respectively. The average recovery was 102.7%±0.5%. The internal standard recovery was 97.9%±2.2%, and the matrix effects were 99.3%±2.8%, 100.6%±3.2% and 99.3%±2.0%, respectively. The average matrix effect was 99.7% ±0.8%. The internal standard matrix effect was 99.3%±3.2%, without hemolysis or hyperlipidemia effects. The in-batch precision (CV %) and accuracy (RE %) of the quality control samples were 1.15%~4.99% and -2.45%~6.93%, respectively, the interbatch precision (CV %) was 2.87%~3.56%, and the interbatch accuracy (RE %) was 1.32%~4.42%. The quality control sample presented good stability at -20°C and -70°C for 2 years, -70°C for repeated freeze- thaw cycles, room temperature and 2~8°C for 3 days, which met the determination requirements.

**Table 6. Incidence of ropivacaine oil delivery depot-related adverse events.**

| Group | Ropivacaine hydrochloride injection | Ropivacaine oil delivery depot | | | | |
|---|---|---|---|---|---|---|
| Dose (mg) | 150 | 150 | 230 | 300 | 350 | 400 |
| n | 3 | 6 | 8 | 8 | 8 | 8 |
| Erythema at injection site | 2 | 4 | 2 | 5 | 1 | 3 |
| Papules at the injection site | 0 | 2 | 0 | 5 | 0 | 0 |
| Hypertriglyceridemia | 1 | 4 | 1 | 1 | 1 | 3 |
| Hypotension | 0 | 1 | 4 | 1 | 2 | 1 |
| Abnormal T wave of ECG | 0 | 0 | 1 | 2 | 0 | 1 |
| Fervescence | 0 | 0 | 0 | 1 | 0 | 1 |
| ECG ST segment depression | 0 | 0 | 0 | 1 | 0 | 0 |
| Abnormal ST segment of ECG | 0 | 0 | 0 | 0 | 0 | 1 |
| Changes in ST-T segment of ECG | 0 | 0 | 0 | 0 | 0 | 1 |
| Total | 3 | 5 | 5 | 7 | 5 | 7 |

ECG, electrocardiogram.

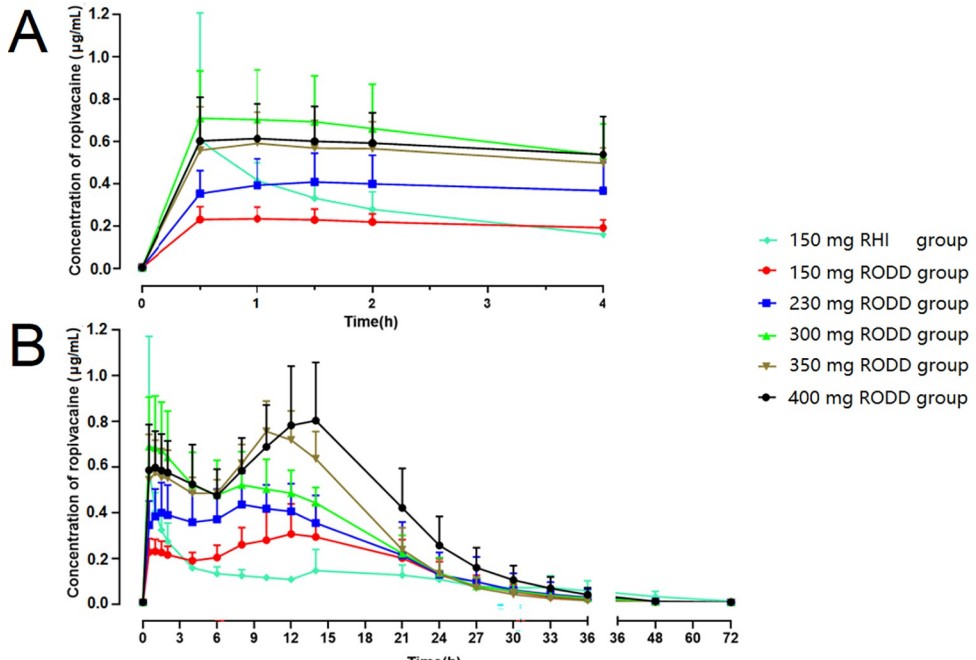

**Fig 7. Curves of ropivacaine concentration-time in plasma after multineedle subcutaneous injection of RODD.**
The results represent the mean±SD (RODD, n = 6~8; RHI, n = 3). After subcutaneous injection of RODD or RHI in healthy subjects' abdomens, venous blood was taken and centrifuged, and the upper layer of plasma was treated with protein precipitation. The content of ropivacaine in plasma was measured at different time points. The concentrations of ropivacaine at 0.5, 1, 1.5 and 2 h after administration are shown in A. The results indicated that ropivacaine could be quickly released from the RODD and enter the blood circulation. The ropivacaine concentration-time curves from 0–72 h are shown in B. Compared with RHI, the concentration of ropivacaine from RODD was higher and showed a bimodal phenomenon.

## Pharmacokinetics

The average plasma concentration-time curve of ropivacaine in each group is presented in Fig 7. The ropivacaine concentration in the RHI group peaked at 0.5 h after administration and then decreased rapidly, suggesting that the release of ropivacaine was rapid. The ropivacaine concentration-time curves in the RODD groups presented double peaks, with the first peak at 1~2 h and the second peak at 10~15 h. After the second peak, the concentration of ropivacaine decreased slowly. The pharmacokinetic parameters are shown in Table 7. The

**Table 7. Pharmacokinetic parameters for ropivacaine oil delivery depot in healthy subjects (mean±SD).**

| Group | Ropivacaine hydrochloride injection | Ropivacaine oil delivery depot | | | | |
|---|---|---|---|---|---|---|
| Dose (mg) | 150 | 150 | 230 | 300 | 350 | 400 |
| n | 3 | 6 | 8 | 8 | 8 | 8 |
| Cmax ($\mu g \cdot mL^{-1}$) | 0.58±0.13 | 0.35±0.09 | 0.50±0.09 | 0.74±0.18 | 0.80±0.12 | 0.86±0.24 |
| AUC0-t ($h \cdot \mu g \cdot mL^{-1}$) | 5.3±0.6 | 6.2±1.9 | 8.6±1.9 | 10.8±2.3 | 12.3±1.7 | 15.3±3.7 |
| AUC0-∞ ($h \cdot \mu g \cdot mL^{-1}$) | 5.4±6.8 | 6.2±1.9 | 8.7±2.0 | 10.85±2.31 | 12.4±1.7 | 15.3±3.7 |
| Tmax (h) | 0.77 ±0.06 | 8.2±5.9 | 10.4 ±5.6 | 3.6± 1.9 | 8.44± 4.68 | 11.8 ±4.6 |
| t1/2 (h) | 7.6 ±5.4 | 3.4 ±0.96 | 3.20 ±1.13 | 3.7± 1.1 | 2.7 ±0.65 | 3.6±0.80 |
| Kel ($h^{-1}$) | 0.17 ±0.18 | 0.22± 0.06 | 0.24 ±0.08 | 0.20 ±0.06 | 0.27±0.06 | 0.20± 0.05 |
| CL ($L \cdot h^{-1}$) | 28.3 ±3.5 | 25.8 ±6.6 | 27.9 ±7.4 | 28.6± 5.5 | 28.8± 4.1 | 27.8± 7.8 |

Cmax, maximum concentration; AUC, area under the curve; Tmax, time of maximum concentration; t1/2, half-life; Kel, elimination rate constant; CL, clearance.

$C_{max}$ in the RHI group (0.58±0.12 μg·mL$^{-1}$), and the $C_{max}$ of the 150 mg RODD group was 0.35±0.09 μg·mL$^{-1}$. The $C_{max}$ values of the 230 and 300 mg RODD groups increased gradually. Moreover, the $C_{max}$ values of the 350 and 400 mg groups were 0.80±0.12 and 0.86 ±0.24 μg·mL$^{-1}$, respectively, and the increases were significant. Additionally, the result of fitting power models showed that the estimated value of ln(dose) regression coefficient of $C_{max}$ and its 90% confidence interval was 0.97 (0.78, 1.16), and it was linearly correlated with an increasing dose (90% decision interval, 0.77~1.23). The $T_{max}$ value of the RHI group was 0.77 ±0.06 h, and the $T_{max}$ of the RODD groups ranged from 3.63±4.98 h to 11.81±4.58 h. The result suggested that $T_{max}$ was delayed as the dose of RODD increased. In addition, $AUC_{0-t}$ and AUC0-∞ were increased as the dose of RODD increased. The average concentration-time of ropivacaine in each group suggested that ropivacaine was continuously released from the RODD into the blood and there were two peaks. Ropivacaine elimination exhibited linear pharmacokinetic characteristics in the terminal curve, namely, the first-order model. This was the same metabolic pattern as that reported for ropivacaine hydrochloride injection [31].

## Onset, duration, and intensity of nerve block

The onset time of nerve block shortened with increasing RODD dose, and the time of the 400 mg group was the fastest, with a median of 0.42 h in all test areas. The duration of the nerve block was dose-independent, and the median of the 400 mg RODD group was 35.7~47.7 h. When the scores of nerve block intensity were greater than 4 (>level 1), the analgesic rate of the 150~400 mg RODD groups were 100%. When the scores of nerve block intensity were greater than or equal to 5 (≥level 2). the analgesic rates of the 150, 230, 300, 350, and 400 mg RODD groups were 13.9%, 22.5%, 32.8%, 30.8%, and 42.9%, respectively, which was positively correlated with the dose of RODD. Finally, in the scores of nerve block intensity greater than and equal to 6 (level 3), the analgesic rate of the RODD groups ranged from 0.99% to 3.6%, a small percentage with no significant dose correlation. Therefore, the highest nerve block intensity was found in the 400 mg RODD group, and the distance between two injection sites on the circumference was 21.5±1.5 mm (Table 8).

**Table 8. Nerve block effects of ropivacaine oil delivery depot.**

| Group | | Ropivacaine oil delivery depot | | | | |
|---|---|---|---|---|---|---|
| Dose (mg) | | 150 | 230 | 300 | 350 | 400 |
| n | | 6 | 8 | 8 | 8 | 8 |
| Onset time (h)[1] | Center of the circle | 0.87 (0.43,8.1) | 0.5 (0.47,0.55) | 0.46 (0.38,0.48) | 0.47 (0.42,0.63) | 0.42 (0.38,0.42) |
| | 1 cm from the center of the circle | 2.4 (0.4,4.2) | 0.5 (0.47,0.55) | 0.46 (0.38,0.48) | 0.47 (0.40,0.63) | 0.42 (0.38,0.47) |
| | 3 cm from the center of the circle | 0.57 (0.45,4.4) | 0.5 (0.46,0.55) | 0.47 (0.42,0.52) | 0.78 (0.40,4.18) | 0.42 (0.38,0.47) |
| | 4 cm from the center of the circle | 0.54 (0.45,1.2) | 0.53 (0.47,4.1) | 0.82 (0.42,4.0) | 1.13 (0.42,4.2) | 0.42 (0.40,1.0) |
| Maintenance time (h)[1] | Center of the circle | 31.2 (9.02,67.9) | 47.5 (22.0,62.0) | 52.7 (35.6,71.6) | 26.57 (20.9,35.7) | 35.73 (26.7,47.9) |
| | 1 cm from the center of the circle | 54.7 (13.7,67.9) | 47.6 (34.9,68.0) | 71.3 (29.7,71.6) | 41.54 (19.9,71.6) | 35.79 (31.9,47.7) |
| | 3 cm from the center of the circle | 37.88 (16.9,71.6) | 47.7 (47.6,71.5) | 33.8 (23.0,47.6) | 43.61 (20.7,67.9) | 47.71 (29.8,71.7) |
| | 4 cm from the center of the circle | 69.5 (23.1,71.6) | 47.7 (35.7,67.9) | 47.7 (34.9,67.9) | 35.68 (29.6,63.9) | 47.43 (26.7,71.7) |
| Nerve block intensity[1] | Level 1: 5>analgesic intensity score>4 | 86.1% | 77.5% | 67.2% | 69.2% | 57.1% |
| | Level 2: 6>analgesic intensity score ≥5 | 10.3% | 19.5% | 29.5% | 29.8% | 40.2% |
| | Level 3: analgesic intensity score ≥6 | 3.6% | 3.0% | 3.30% | 0.99% | 2.7% |
| Distance between two injection sites (mm)[2] | | 40.9±1.3 | 35.2±2.2 | 29.6±2.3 | 24.0±1.9 | 21.5±1.5 |

Note

[1], Median, 95%, CI; Note

[2], mean±SD.

## Discussion

An injectable oil delivery depot (ODD) is a classic oily drug delivery system that is a vehicle for hormone or psychotropic drugs. Previous clinical studies have examined the application of ODD in the field of anesthesia, such as the formulation of ropivacaine with castor oil and phospholipids. However, the high levels of irritation and sensitization associated with castor oil are causes for concern [32, 33]. In contrast, the RODD formulation that we designed for clinical trials, with high concentrations of ropivacaine has a stronger nerve block analgesic effect. Benzyl alcohol is a rapid analgesic for common injections. Benzyl benzoate has an anti-inflammatory effect, and soybean oil is the main excipient for lipid emulsion for injection [34–36]. Therefore, RODD was judged to have better clinical safety and anesthetic effects.

RODD was allowed for further clinical studies in healthy subjects after evaluation in animal studies. In exploratory studies, there was slight local irritation at a single infiltration dose of RODD 24 mg. Local irritation slightly increased after multineedle subcutaneous administration. In most cases, it was relieved after 24 h and disappeared after 72 h. It was speculated that the local irritation might be related to the overlapping infiltration range of two adjacent needles. The incidence of RODD-related adverse events was mild, including hypotension, abnormal ST-segment electrocardiogram, and hypertriglyceridemia, which returned to normal 72 h after administration. In addition, two subjects in the RHI group had mild erythema in the administered area, and one subject had hypertriglyceridemia, which quickly disappeared. These transient and mild adverse reactions may be caused by ropivacaine, excipients, frequent clinical administration, or topical disinfectants. In addition, no abnormal mental reaction was observed. Therefore, RODD was demonstrated to be safe. The results support the follow-up study of subcutaneous administration of analgesia in patients.

The preliminary pharmacodynamic assessment of RODD was divided into two parts. The recommended amount of RODD in single-needle administration was 24 mg in the exploratory experiment, the duration of analgesia was approximately 12 h, and the injection range was approximately 40 mm. The circle on the abdomen of the subject was uniformly administered with 6 needles in the 400 mg RODD group, the distance between two adjacent needles was 20 mm, and the subcutaneous injection range overlapped each other, suggesting that the anesthesia time of RODD multipoint injection was further prolonged. The efficacy of local infiltration of RHI has been reported [37–39]. The analgesic onset time and duration of RHI were approximately 10~45 min and 6~8 h, respectively. Compared with RHI, the analgesic onset time of the 400 mg RODD group was similar, and the duration of analgesia was increased by more than 6-fold (vs. 35.7~47.7 h, median), which could meet the needs of postoperative analgesia. Therefore, the analgesic effect of RODD was significantly prolonged, and the potential clinical advantages were obvious.

The concentration-time curves of ropivacaine for RODD were significantly different from those for RHI. The RHI curve rose rapidly after administration, and the curve decreased rapidly after 0.77 h ($T_{max}$) and was then prolonged at a lower concentration. The first peak (589.4 ±158.9 ng·mL$^{-1}$) of the curve in the RODD 400 mg group was reached at 1 h after administration, and the second peak (864.9±237.5 ng·mL$^{-1}$) was reached at 11.8 h ($T_{max}$). Then, the curve decreased slowly. The curves of the 150~350 mg RODD group were similar to those of the 400 mg RODD group. This phenomenon has also been found in preclinical animal studies of RODD, including pharmacokinetics in dogs and toxicokinetics in rats [26, 40]. This bimodal phenomenon has not been found with other clinically used oil preparations. The metabolic pathway of ropivacaine in the body is clear, and ropivacaine has no hepato-enteric ciculation [31]. Therefore, we hypothesized that the bimodal ropivacaine-time curve characteristic of RODD in subcutaneous injection was caused by the continuous release of ropivacaine, which

might be related to the combined effect of free ropivacaine and ODD. The ropivacaine outside RODD releases to produce the first peak. During ODD metabolism, ropivacaine in RODD is rereleased to generate the second peak.

Pharmacokinetic studies of RODD showed that the $T_{max}$ of the RHI 150 mg group was lower than 1 h, while that of the RODD 150~400 mg groups lagged 3.6~11.8 h, suggesting that ropivacaine can be released slowly. Compared with the 150 mg RHI group, the $C_{max}$ value in RODD 150 mg group decreased by 40%, and the value increased by 27% and 48% when the doses were increased to 350 mg and 400 mg, respectively. This value is much lower than the reported $C_{max}$ of ropivacaine hydrochloride injection at the maximum tolerated dose (2.9 ±0.5μg·mL$^{-1}$) [41]. All of the above findings suggest that RODD was safe. Second, the elimination half-life ($t_{1/2}$) of the 150 mg RHI group was 7.6±5.4 h, which was higher than that of the 150 mg RODD group. The reason might be the small number of subjects in the RHI group or the difference in ropivacaine concentration between the RODD and RHI groups.

Compared with the report of RHI [19, 20], the dose of ropivacaine in the clinical application of RODD was doubled, a single administration of 400 mg, the duration of anesthesia was extended by approximately 4 times, and the duration of efficacy was more than 35 h. In healthy subjects, adverse events were mild and quickly resolved after subcutaneous abdominal injection of RODD. The results suggest that RODD is of great significance for postoperative analgesia. However, the application of RODD in clinical patients still needs to be further evaluated and explored.

The results of the preclinical safety evaluation demonstrated that RODD does not interfere with incision healing in the abdomen and back of mini-pigs. However, RODD was not used to infiltrate the patient's incision after surgery, and the safety and effect of RODD need to be verified by experiments, including the dose of RODD in patients with incision infiltration and the distance between the two injection sites, as well as the distance from the injection site to the incision, which should be studied; whether the injection of RODD affects the wound healing of patients after surgery and the occurrence of related adverse events; whether the duration of analgesia is similar to that of healthy subjects; and whether RODD is used for surgical incisions in other parts of the body except the abdomen. How strong is the analgesic effect? At present, we are designing a program to evaluate the analgesic effect and safety of endoscopic incision of RODD in the abdomen of patients, and these indicators have been given priority consideration.

The release mechanism in ODD has also been studied. For instance, Kalicharan et al. reported the metabolism of ODD in different tissues and measured its spatial distribution by magnetic resonance imaging technology to visualize the changes in ODD *in vivo*. They found that 1 mL ODD was injected into the vastus lateralis and deltoid muscles in healthy subjects, and the ODD could be cleared from the body within a week [42, 43]. Our study suggested that the median duration of analgesia in the 400 mg RODD group was approximately 2 days. Two pathways have been hypothesized. Ropivacaine is released synchronously with ODD and is maintained at local effective concentrations for approximately 2 days and at lower concentrations for approximately 5 days. Alternatively, ropivacaine may be released more quickly than oil-based excipients. Therefore, further *in vivo* research needs to examine the mechanism of ropivacaine release from RODD, and we will report it in the future.

## Conclusions

RODD has a sustained release effect with a delayed $T_{max}$ and a long duration of nerve block. No abnormal reaction was found in the RODD containing 400 mg of ropivacaine after subcutaneous injection among healthy subjects, suggesting that RODD was adequately safe.

## Supporting information

**S1 Checklist. CONSORT 2010 checklist.**
(DOC)

**S1 File. Study protocol (Brief English version).**
(PDF)

**S2 File. Study protocol (Original Chinese version).**
(PDF)

## Acknowledgments

The authors are grateful to all the subjects who took part in the trial, as well as the team of doctors team in the Department of Anesthesiology and the Phase I Clinical Research Institution at You'an Hospital, Capital Medical University.

## Author Contributions

**Conceptualization:** Wu-dang Lu, Yong-xiao Cao.

**Formal analysis:** Wu-dang Lu, Mei-xia Wang, Bo Yang.

**Investigation:** Wu-dang Lu, Dan-ling Zhao, Mei-xia Wang, Ya-qi Jiao, Min Zhang, Jian-ping Dong, Hai-bo Zhang, Yi Yang, Ye Tian, Min-quan Hui, Bo Yang.

**Methodology:** Dan-ling Zhao, Ya-qi Jiao, Ping Chi, Bo Ma, Yong-xiao Cao.

**Project administration:** Wu-dang Lu, Dan-ling Zhao, Mei-xia Wang, Ya-qi Jiao, Ping Chi, Min Zhang, Bo Ma, Jian-ping Dong, Hai-bo Zhang, Yong-xiao Cao.

**Resources:** Dan-ling Zhao, Mei-xia Wang, Ping Chi, Min Zhang, Yi Yang.

**Validation:** Hai-bo Zhang, Ye Tian.

**Writing – original draft:** Wu-dang Lu.

**Writing – review & editing:** Wu-dang Lu, Yi Yang, Yong-xiao Cao.

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
