## [Decision Letter · Decision Letter 0]

24 Aug 2022

PONE-D-22-07472

A randomized, single-blind study of ropivacaine oily delivery depot: safety, pharmacokinetics and preliminary pharmacodynamics of subcutaneous infiltration in healthy subjects

PLOS ONE

Dear Dr. Cao,

Thank you for submitting your manuscript to PLOS ONE. After careful consideration, we feel that it has merit but does not fully meet PLOS ONE’s publication criteria as it currently stands. Therefore, we invite you to submit a revised version of the manuscript that addresses the points raised during the review process.

Your manuscript has been reviewed by two peer-reviewers and their reports are appended below. 

The reviewers comment that essential parts of the methodology, such as randomisation procedures, sample size calculations, API characterisitcs, GMP manufacturing process, etc, are missing or require further detail and/or clarification. The reviewers also comment that the statistical power of the study needs to be assessed more carefully. Furthermore, the reviewers comment that the results section does not report on essential aspects of the study, and that the discussion is incomplete. 

In addition to the reviewers' comments, it has come to the editor's attention that the authors have declared "no competing interests" despite multiple authors being affiliated to pharmaceutical companies. Please carefully review PLOS ONE's Competing Interests policy. Please note that failure to declare competing interests can result in immediate rejection of a manuscript. If an undisclosed competing interest comes to light after publication, PLOS will take action in accordance with COPE guidelines and issue a public notification (Correction, Expression of Concern, or Retraction) to the community.

We look forward to receiving your revised manuscript.

Kind regards,

Maria Elisabeth Johanna Zalm, Ph.D

Editorial Office

PLOS ONE

https://journals.plos.org/plosone/s/file?id=ba62/PLOSOne_formatting_sample_title_authors_affiliations.pdf".

Reviewers' comments:

Reviewer's Responses to Questions

**Comments to the Author**

1. Is the manuscript technically sound, and do the data support the conclusions?

Reviewer #1: Yes

Reviewer #2: Partly

2. Has the statistical analysis been performed appropriately and rigorously? 

Reviewer #1: Yes

Reviewer #2: No

3. Have the authors made all data underlying the findings in their manuscript fully available?

Reviewer #1: Yes

Reviewer #2: Yes

4. Is the manuscript presented in an intelligible fashion and written in standard English?

Reviewer #1: Yes

Reviewer #2: No

5. Review Comments to the Author

Reviewer #1: Interesting paper, although with relevant flaws from a methodological point of view

Some issues

Abstract. Why 6 groups? please clarify better

Abstract. Primary end point should be better detailed

Abstract, results. They should be rewritten in a clearer way, from primary to secondary end points

Methods. Inclusion/exclusion criteria should be better detailed. Were all patients consecutive?

Methods: Randomization procedures are totally lacking and should be added

Methods: Was blinding (of patients, physicians, and outcome assessors) used? If not, do the authors acknowledge

possible biases resulting from a lack of blinding?

Methods. sample size calculation is missing, as details about primary and secondary end points

Methods. Did authors perform any difference towards protocol?

Methods. Due to small sample size, normal distribution should be checked

Reviewer #2: The manuscript from Wu-dang Lu and co-authors deals with a clinical trial in healthy patients of an oily depot formulation of ropivacaine.

There are several concerns about the methods used and the way in which the results are interpreted to be addressed before considering this manuscript suitable for publication:

1. The literature on the available lipid-based depots available is not up to date and the articles cited are simply presented as a list of studies without being harmonised and critically presented to support the study carried out by the authors. It is recommended to improve this section and making it more meaningful for the readers.

2. The material section is completely absent. This is a clinical trial and no information about the API's characteristics (supplier, batch, chemical form, purity) is provided. No information about the RHI used in the control group is given. No information about the GMP manufacturing process (was a GMP manufacturing at all?) or on any pharmacopoeial test on parenteral administration was provided. Not even information on the vehicle is given! It is also not mentioned how the concentration 20 mg/mL was assessed and how the 2 years shelf life was determined (not mentioned in the manuscript and not even references to already published methods are given).

3. In the ethics statement the recruitment criteria (inclusion/exclusion) are not given. It is not even mentioned whether the biological sex was considered, and it seems it was not considered in the analysis. For analgesics (particularly when the pharmacodynamics is investigated) it should be integrated in the results and discussion section. It could be helpful in this section to refer to Table 3 for the criteria analysed.

4. In the trial design and subject it is unclear why only 3 patients were injected with RHI. This is a pivotal control. The statistical power of the study must have suffered because of this low number of patients with respect to the other groups. Could the authors comment on that?

5. Study 1: It is unclear how many of the random directions of the infiltration points were selected. Were the healthy subjects injected multiple times to test all the random directions or only once? If multiple times, was a "washout" time respected? This section should be rephrased and make it clearer to the reader, possibly moving part of the figure caption into the main text.

6. Study 2: as previously commented, here n=3 for RHI seems to be a very limited number of subjects to have solid statistical analysis. In this regard: almost no information on the type of statistical analysis run is provided, although in several points of the manuscript it is mentioned that some data are significantly different from others. Asterisks are even absent from the plots and this makes impossible to assess the quality of the analysis and the robustness of the presented data. Another essential control that is missing is the injection of the vehicle alone.

7. In the PK section it is reported that proteins in plasma samples were precipitated after freezing, just prior to analysis. Why the authors did not opt to precipitate proteins before freezing? The validation of the API extraction method is not reported. This information should be provided, and if the method was previously validated a reference should be provided. Furthermore, some more technical details (such as the type of tubes used for the sample preparation: low protein binding? plastic chose, supplier, volume) should be given.

8. The preliminary pharmacodynamics is based on measuring "acupuncture on circle 1 and 2 of the abdomen". No other detail is provided, compromising the possibility of reproducing these results.

9. Table 6 should be improved, it is unclear from the labelling of the table column what the numbers refer to.

10. The Safety results need to be revised: the results on RHI are not discussed.

11. The Pharmacokinetics results need to be revised: it is not described which PK model was used; it is not specified how Cmax was determined when the profiles had two peaks; how the stats were run is not described; the Tmax of the 300 mg dose is much lower than the other depots (3 h) and this information is not presented in the results and not discussed later; the t1/2 of ropivacaine is strongly decreased in the depots and this information is not presented and not discussed.

12. In the PD section, it is stated that the duration of the nerve block is dose-independent, but from the table level 2 is clearly dose-dependent. The authors are recommended to revise this section and comment the table more extensively.

13. As stated in my comment 6, the vehicle without ropivacaine should have been administered, especially considering that in the Discussion the authors claim a possible synergistic effect of the excipients that were selected for this study.

14. Explanations on the two peaks of ropivacaine in plasma for the depot formulations are not given but they should. What are the hypotheses of the authors for this behaviour? This is usually not a wished effect for depot formulations. The Tax for the RODD is given as between 2.4 and 11 h, but the lowest value should be 3.4 of the 300 mg dose and 11.8 of the 400 mg.

15. The final part of the discussion is incomplete and it is not conclusive. The authors are encouraged to re-write it. The conclusions are minimalistic, with just one sentence.

Several typos are present throughout the manuscript. Below some examples, but the authors are strongly encouraged to proofread the text again before re-submission.

Page 4 "low lipid soluble", "introperitoneally"

Page 6 "cilin"

Page 7 "china"

Page 12 "vortices"

Page 13 "summerizeda"

Page 14 "Health subjects"

Page 19, "RODD is safety" (twice in this page)

Several international style guides recommended to leave a space between the numbers and their unit of measurements. This style is inconsistent throughout the document and should be re-checked before submission.

Last but not least, there is no disclosure concerning the competing interests but among the affiliations two companies are listed.

6. PLOS authors have the option to publish the peer review history of their article (what does this mean?). If published, this will include your full peer review and any attached files.

Reviewer #1: **Yes: **Fabrizio D'Ascenzo

Reviewer #2: No

---

## [Author Response · Author response to Decision Letter 0]

27 Oct 2022

Dear Dr. Maria Elisabeth Johanna Zalm, 

Thank you for your kind consideration for evaluation of our manuscript PONE-D-22- 07472, entitled “A randomized trial: the safety, pharmacokinetics and preliminary pharmacodynamics of ropivacaine oily delivery depot in healthy subjects”. Based on your comments and suggestions, we have modified our manuscript and responsed reviewers’ comments point-by-point showed in the following table (attachment 1). Meanwhile, a revised manuscript is also attached. The manuscript has been further polished and the certificate is showed the attachment 2.

In addition, the reviewers had confirmed that all the raw data were fully available at the first review of the manuscript. We have reviewed the raw data again. It had been uploaded to a public repository "ResMan" (www.Medresman.org.cn. ‘ResMan’ was associated with the Chinese Clinical Registry (ChiCTR)). 

Supporting information, acknowledgments and author contributions were added to the manuscript. The attachments were added, including updated CONSORT checklist, chinese original protocol and english brief protocol.

We hope the revised version will meet your requirements. 

We are looking forward to your reply.

Best Regards,

Yong-xiao Cao

Department of Pharmacology, 

School of Basic Medical Sciences, 

Xi’an Jiaotong University Health Science Center

yxy@xjtu.edu.cn

---

## [Decision Letter · Decision Letter 1]

28 Apr 2023

PONE-D-22-07472R1A randomized trial: the safety, pharmacokinetics and preliminary pharmacodynamics of ropivacaine oily delivery depot in healthy subjectsPLOS ONE

Dear Dr. Cao,

Thank you for submitting your manuscript to PLOS ONE. After careful consideration, we feel that it has merit but does not fully meet PLOS ONE’s publication criteria as it currently stands. Therefore, we invite you to submit a revised version of the manuscript that addresses the points raised during the review process.

After a careful reading of your manuscript we believed that it can be submitted for publication in Plos One. To attend the scientific criteria of the Journal we invite you to proceed changes in the manuscript according to the reviewers #3 and #4 comments Best regardsJosé Luiz Vieira 

Please submit your revised manuscript by Jun 12 2023 11:59PM. If you will need more time than this to complete your revisions, please reply to this message or contact the journal office at plosone@plos.org. Please include the following items when submitting your revised manuscript:A rebuttal letter that responds to each point raised by the academic editor and reviewer(s). You should upload this letter as a separate file labeled 'Response to Reviewers'.A marked-up copy of your manuscript that highlights changes made to the original version. You should upload this as a separate file labeled 'Revised Manuscript with Track Changes'.An unmarked version of your revised paper without tracked changes. You should upload this as a separate file labeled 'Manuscript'.If applicable, we recommend that you deposit your laboratory protocols in protocols.io to enhance the reproducibility of your results. Protocols.io assigns your protocol its own identifier (DOI) so that it can be cited independently in the future. For instructions see: https://journals.plos.org/plosone/s/submission-guidelines#loc-laboratory-protocols. Additionally, PLOS ONE offers an option for publishing peer-reviewed Lab Protocol articles, which describe protocols hosted on protocols.io. Read more information on sharing protocols at https://plos.org/protocols?utm_medium=editorial-email&utm_source=authorletters&utm_campaign=protocols.

We look forward to receiving your revised manuscript.

Kind regards,

José Luiz Fernandes Vieira

Academic Editor

PLOS ONE

Journal Requirements:

Reviewers' comments:

Reviewer's Responses to Questions

**Comments to the Author**

1. If the authors have adequately addressed your comments raised in a previous round of review and you feel that this manuscript is now acceptable for publication, you may indicate that here to bypass the “Comments to the Author” section, enter your conflict of interest statement in the “Confidential to Editor” section, and submit your "Accept" recommendation.

Reviewer #1: All comments have been addressed

Reviewer #2: (No Response)

Reviewer #3: (No Response)

Reviewer #4: (No Response)

2. Is the manuscript technically sound, and do the data support the conclusions?

Reviewer #1: (No Response)

Reviewer #2: Partly

Reviewer #3: Yes

Reviewer #4: Partly

3. Has the statistical analysis been performed appropriately and rigorously? 

Reviewer #1: (No Response)

Reviewer #2: No

Reviewer #3: Yes

Reviewer #4: Yes

4. Have the authors made all data underlying the findings in their manuscript fully available?

Reviewer #1: (No Response)

Reviewer #2: Yes

Reviewer #3: Yes

Reviewer #4: Yes

5. Is the manuscript presented in an intelligible fashion and written in standard English?

Reviewer #1: (No Response)

Reviewer #2: Yes

Reviewer #3: Yes

Reviewer #4: No

6. Review Comments to the Author

Reviewer #1: (No Response)

Reviewer #2: The authors are kindly requested to provide a point-to-point reply. The revised manuscript with no track change has been uploaded, together with a version with a full-mode track-change with all the edits, but the changes have not been explained nor commented and it is extremely difficult to understand which points where addressed and which not.

For example, the PK section has been only partially modified.

No information on the statistical analysis used is given and still the data are commented with statistically significant differences.

A conflict of interests section is still missing.

Authors' names in the reference list should be double-checked (e.g. ref 33 Di Berardino L, not Di BL, etc)

Reviewer #3: 1. Suggest to add the annotate for Fig A and B in Fig. 7;

2. It was shown in Fig. 7 B that the second peak concentration in 400mg group was achieved at 14 hours after administration, while the subsequent sampling time was 21 hours after administration. It was possible that the sampling point setting was unreasonable, resulting in the loss of true peak concentration and the AUC;

3. In the discussion part, the authors mentioned that Kalicharan et al. reported 1mL ODD into the vastus lateralis and deltoid muscles could be cleared from the body within a week. While the result of this study showed the median duration of analgesia in RODD 400 mg group was 35.7~47.7 h. Please add further discussion on the relationship between the retention time of ODD and the analgesic effect of RODD.

4. The discussion and the conclusion should be focused on the purpose of the study and the clinical value.

Reviewer #4: The study and its outcomes seem important to publish. The manuscript itself needs some work to make the methods and conclusions clear and understandable. I suggest a major rewrite to bring the language within standard English expectations before publication.

7. PLOS authors have the option to publish the peer review history of their article (what does this mean?). If published, this will include your full peer review and any attached files.

Reviewer #1: **Yes: **Fabrizio D'Ascenzo

Reviewer #2: No

Reviewer #3: **Yes: **Jia Miao

Reviewer #4: No

---

## [Decision Letter · Decision Letter 2]

7 Sep 2023

A randomized trial: The safety, pharmacokinetics and preliminary pharmacodynamics of ropivacaine oil delivery depot in healthy subjects

PONE-D-22-07472R2

Dear Dr. Cao

After After a careful evaluation of the rebuttal letter to the reviewer, we found that all the questions posed by the reviewer were well answered by the authors. The most important points of the review were clarified and the references wetre adjusted. Thus, we recommend the approval of the manuscript for publication in PLOS ONE.

Best regards

José Luiz Vieira

We’re pleased to inform you that your manuscript has been judged scientifically suitable for publication and will be formally accepted for publication once it meets all outstanding technical requirements.

Kind regards,

José Luiz Fernandes Vieira

Academic Editor

PLOS ONE

Additional Editor Comments (optional):

Reviewers' comments:

Reviewer's Responses to Questions

**Comments to the Author**

1. If the authors have adequately addressed your comments raised in a previous round of review and you feel that this manuscript is now acceptable for publication, you may indicate that here to bypass the “Comments to the Author” section, enter your conflict of interest statement in the “Confidential to Editor” section, and submit your "Accept" recommendation.

Reviewer #4: (No Response)

2. Is the manuscript technically sound, and do the data support the conclusions?

Reviewer #4: Yes

3. Has the statistical analysis been performed appropriately and rigorously? 

Reviewer #4: Yes

4. Have the authors made all data underlying the findings in their manuscript fully available?

Reviewer #4: Yes

5. Is the manuscript presented in an intelligible fashion and written in standard English?

Reviewer #4: Yes

6. Review Comments to the Author

Reviewer #4: Thank you for addressing some of my original comments. It is extremely important to provide a response to the reviewers for each comment as there were some comments made that it wasn't clear if they were addressed or not. The manuscript reads much better than before. Below are some outstanding comments:

1) Just an overall general comment to please provide a response to each comment made by each reviewer. It is extremely important as it helps reviewers understand why their comments were addressed as such or why they were not addressed.

2) In the methods section the sentence "The primary endpoint was..." is still confusing. A dose cannot be an endpoint but rather a response to a dose (i.e., safety, efficacy) may be an endpoint. Please be very clear what your endpoint was. It appears to me that the endpoint was to find a safe dose not to exceed a maximum of 400 mg of ropivacaine in a ODD formulation. If so, please state this very clearly so that the reader knows exactly what the purpose of the study was.

3) In the third paragraph of the Introduction it is stated that an analgesic duration less than 8 hr cannot meet the needs of posoperative analgesia. Suggest changing the word "cannot" to "likely will not".

4) In the optimal dose of single-needle injection of RODD section it is stated that the nerve block range and areas under the curve for RODD were "obviously" higher than those of the 12, 16 and 20 mg RHI groups. I suggest you remove the word "obviously" as it is subjective and should not be used in manuscripts.

5) In the discussion section the word "cuases" is misspelled and should be "causes"

7. PLOS authors have the option to publish the peer review history of their article (what does this mean?). If published, this will include your full peer review and any attached files.

Reviewer #4: No

---

## [Editor Report · Acceptance letter]

11 Sep 2023

PONE-D-22-07472R2 

A randomized trial: The safety, pharmacokinetics and preliminary pharmacodynamics of ropivacaine oil delivery depot in healthy subjects 

Dear Dr. Cao:

I'm pleased to inform you that your manuscript has been deemed suitable for publication in PLOS ONE. Congratulations! Your manuscript is now with our production department. 

Kind regards, 

on behalf of

Dr. José Luiz Fernandes Vieira 

Academic Editor

PLOS ONE